# Oligometastasis of Gastric Cancer: A Review

**DOI:** 10.3390/cancers16030673

**Published:** 2024-02-05

**Authors:** Itaru Yasufuku, Hiroshi Tsuchiya, Seito Fujibayashi, Naoki Okumura, Yuki Sengoku, Masahiro Fukada, Ryuichi Asai, Yuta Sato, Jesse Yu Tajima, Shigeru Kiyama, Takazumi Kato, Yoshihiro Tanaka, Katsutoshi Murase, Nobuhisa Matsuhashi

**Affiliations:** 1Department of Clinical Anatomy Development Studies, Gifu University Graduate School of Medicine, Yanagito 1-1, Gifu City 501-1194, Japan; yasufuku.itaru.k5@f.gifu-u.ac.jp; 2Department of Gastroenterological and Pediatric Surgery, Gifu University Graduate School of Medicine, Yanagito 1-1, Gifu City 501-1194, Japan; tsuchiya.hiroshi.m9@f.gifu-u.ac.jp (H.T.); fujibayashi.seito.j6@f.gifu-u.ac.jp (S.F.); okumura.naoki.r7@f.gifu-u.ac.jp (N.O.); sengoku.yuki.w9@f.gifu-u.ac.jp (Y.S.); hukada.masahiro.u9@f.gifu-u.ac.jp (M.F.); asai.ryuichi.k0@f.gifu-u.ac.jp (R.A.); tajima.jesse.yu.s4@f.gifu-u.ac.jp (J.Y.T.); kiyama.shigeru.x8@f.gifu-u.ac.jp (S.K.); kato.takazumi.v5@f.gifu-u.ac.jp (T.K.); tanaka.yoshihiro.h5@f.gifu-u.ac.jp (Y.T.); murase.katsutoshi.f1@f.gifu-u.ac.jp (K.M.)

**Keywords:** oligo-recurrence, oligometastasis, systemic therapy, local therapy, surgery, multimodal therapy, perioperative chemotherapy, distant metastasis, gastric cancer, gastroesophageal adenocarcinoma

## Abstract

**Simple Summary:**

Oligometastatic cancer describes an intermediate stage of cancer between localized and widely spread disease. Oligometastasis can be treated with local treatments, such as surgical removal or radiotherapy. The concept of oligometastasis is also observed in the field of gastric cancer. However, the definition of oligometastasis in gastric cancer is not yet established. Metastatic lesions that are localized and technically removable at diagnosis and that demonstrate a certain response to chemotherapy sometimes present a favorable survival outcome with local treatment. Those metastatic lesions are considered oligometastasis in the field of gastric cancer. Major guidelines for gastric cancer describe the concept of oligometastasis, and a European multicenter board of experts organized the OligoMetastatic Esophagogastric Cancer Consortium to establish the definition, diagnosis, and treatment for oligometastasis. Two large phase 3 studies are ongoing to evaluate the survival outcomes of patients with oligometastasis treated with perioperative chemotherapy combined with surgical removal.

**Abstract:**

The concept of oligometastasis is not yet fully established in the field of gastric cancer. However, metastatic lesions that are localized, technically resectable at diagnosis, present a certain response to preoperative chemotherapy, and present favorable survival outcomes with local treatments, sometimes in combination with chemotherapy, are recognized as oligometastasis in the field of gastric cancer. Oligometastasis is noted in European Society for Medical Oncology guidelines and Japanese gastric cancer treatment guidelines, and local treatment is mentioned as one of the pivotal treatment options for oligometastasis. Solitary liver metastasis or a small number of liver metastases; retroperitoneal lymph node metastasis, especially localized para-aortic lymph node metastasis; localized peritoneal dissemination; and Krukenberg tumor are representative types of oligometastasis in gastric cancer. The AIO-FLOT3 trial prospectively evaluated the efficacy of multimodal treatments for gastric cancer with oligometastasis, including surgical resection of primary and metastatic lesions combined with chemotherapy, confirming favorable survival outcomes. Two phase 3 studies are ongoing to investigate the efficacy of surgical resection combined with perioperative chemotherapy compared with palliative chemotherapy. Thus far, the evidence suggests that multimodal treatment for oligometastasis of gastric cancer is promising.

## 1. Introduction

Gastric cancer with distant metastasis is commonly recognized as unresectable, and the standard treatment for patients with this condition is palliative chemotherapy [1,2,3]. The median overall survival (OS) time of patients undergoing palliative chemotherapy for gastric cancer with distant metastasis ranged from around 9.3 to 13.0 months in clinical trials 20 years ago [4,5]. However, recent developments in the availability of new agents and chemotherapeutic regimens have prolonged the survival time of patients with gastric cancer with distant metastasis [6,7,8,9,10,11,12,13]. The median OS time reached 18 months in the SPOTLIGHT trial [13], which investigated the efficacy of first-line zolbetuximab plus 5-fluorouracil, leucovorin, and oxaliplatin (mFOLFOX6) in patients with claudin 18.2-positive, human epidermal growth factor receptor 2 (HER2)-negative, unresectable metastatic gastric cancer or gastroesophageal junction adenocarcinoma.

In addition to the improvement in OS, the evolution of chemotherapy has produced benefits in terms of tumor response and disease control. The CheckMate 649 trial [7] is a phase 3 study that investigated the efficacy of nivolumab combined with chemotherapy as a first-line treatment for advanced gastric and gastroesophageal cancer. The overall response rate was 60% in the nivolumab plus chemotherapy group compared with 45% in the chemotherapy group, and the proportion of patients with a complete response to each regimen was 12% and 7%, respectively. Recent developments in chemotherapy have achieved remarkable tumor shrinkage and even disappearance of metastatic lesions [7,8]. There have also been some cases in which distant metastasis could be remarkably controlled during chemotherapy. Encouraged by the recent developments in chemotherapy, local treatment by surgical resection has been re-considered as an option for patients with relatively limited distant metastasis or patients in which distant metastasis has been controlled with palliative chemotherapy.

Oligometastasis was originally reported in 1995, and it is defined as the state between localized disease and systemic metastasis [14]. It has been reported that local treatments, such as surgical resection or radiotherapy, may prolong the survival time of patients with oligometastasis [14]. For cancers in which the recurrent tumor is the object of local treatment, metachronous metastasis is also included in the definition of oligometastasis as oligo-recurrence [15]. For lung cancer, the prognostic factors of oligometastasis have been evaluated, and metachronous metastasis (or oligo-recurrence) has demonstrated a better prognosis than synchronous oligometastasis for patients undergoing stereotactic body radiation therapy [16]. The differences in survival outcomes between synchronous oligometastasis and metachronous oligo-recurrence may be explained using the tumor activity of the primary lesion. Synchronous oligometastasis is accompanied by the primary tumor. For patients with untreated primary lesions, cases with adjacent organ involvement or metastasis to surrounding lesional lymph nodes from the primary lesion were sometimes included. The malignancy of the primary lesion might influence the poor survival outcomes [16]. In patients with oligo-recurrence, the primary lesion was resected or well-controlled. Localized recurrence observed a long time after treatment for primary disease may suggest that the disease may be more localized in patients with oligo-recurrence than in patients with synchronous oligometastasis. Recently, the concept of oligometastasis has extended to various sites of primary cancer, and the efficacy of local treatment compared with standard palliative treatment for oligometastasis has been evaluated in a randomized phase 2 trial [17]. In this trial, patients with various types of primary cancer were recruited, and the inclusion criteria for oligometastasis were metastasis with a maximum of three lesions in one organ and five metastatic lesions in total. The types of primary cancer included colorectal, breast, lung, and prostate cancer, as well as some other solid tumors. The metastatic sites of oligometastasis were the lungs, bone, liver, adrenal glands, brain, lymph nodes, and pararenal sites. Stereotactic ablative radiotherapy (SABR) was applied as a local treatment for oligometastasis in this trial. The trial adopted a randomized phase 2 screening design, with a two-sided α of 0.20, and the primary endpoint was the OS time. Ninety-nine patients were recruited, and the OS time was 41 months (95% confidence interval [CI] 26 months–upper limit not reached) for patients who underwent SABR and 28 months (95% CI 19–33 months) for patients who underwent standard palliative care (hazard ratio [HR] 0.57, 95% CI 0.30–1.10, *p* = 0.090). The study concluded that local treatment was associated with an improvement in OS time.

For gastric cancer, distant metastasis and recurrence after surgery have been recognized as poor prognostic factors, and surgical resection is not often used in such cases. The REGATTA trial [18], which investigated the efficacy of palliative gastrectomy of the primary tumor prior to palliative chemotherapy in patients with advanced gastric cancer, did not demonstrate the superiority of palliative gastrectomy over palliative chemotherapy alone, and the indication for surgery in these patients is limited. However, a recent survey revealed favorable survival outcomes for patients with limited distant metastasis, such as solitary liver metastasis, a small number of liver metastases, or para-aortic lymph node metastasis, who underwent surgical resection of metastatic lesions [19,20,21,22]. This restoration of surgical resection for distant metastasis from gastric cancer has been achieved by appropriately selecting patients in whom the metastatic site is localized and controlled with perioperative chemotherapy. Thus, localized metastasis or metastatic disease with a slight tumor burden for which multimodal treatments, including surgery combined with chemotherapy, are practically applied have come to be recognized as oligometastasis in the field of gastric cancer, and the number of reports concerning the treatment results of oligometastasis has increased.

Herein, we provide an overview of the current concept and treatment outcomes for oligometastasis in patients with gastric cancer.

## 2. Description of Oligometastasis in Gastric Cancer Guidelines

The National Comprehensive Cancer Network guidelines [1] do not include topics concerning oligometastasis or surgical resection for limited distant metastasis. In the European Society for Medical Oncology (ESMO) guidelines [2], although gastrectomy is not recommended for metastatic disease and there is no obvious definition of oligometastasis, resection of distant metastasis might be considered as an individualized approach in highly selected cases with oligometastatic disease who respond to chemotherapy [2]. In this guideline, the AIO-FLOT3 trial [23] is considered a trial that investigated the treatment result of surgery combined with perioperative chemotherapy for oligometastasis. The AIO-FLOT3 trial was a phase 2 clinical trial evaluating the efficacy and safety of surgical resection combined with perioperative fluorouracil, leucovorin, oxaliplatin, and docetaxel (FLOT) therapy for resectable or limited metastatic gastric and gastroesophageal junction cancer [23]. The trial included patients with resectable metastasis, such as retroperitoneal lymph node metastasis and/or technically resectable metastasis localized to one organ, such as peritoneal metastasis that had not extended to the distant peritoneum, liver metastasis with fewer than five lesions, Krukenberg tumor, adrenal gland metastasis, and extra-abdominal lymph node metastasis. Those types of metastasis were consistent with oligometastasis and were included as one of three groups (Arm B), in which patients were treated with perioperative FLOT therapy combined with surgical resection of the primary tumor and metastatic lesions (if these remained after preoperative chemotherapy). The other groups included patients with resectable non-metastatic disease who underwent surgery combined with perioperative FLOT therapy (Arm A) and patients with metastatic disease who did not fulfill the criteria for Arm B but who were treated with FLOT therapy (Arm C). Thirty-six of 60 patients (60%) in Arm B proceeded to undergo surgery after preoperative chemotherapy, and the OS time of all patients in Arm B was 22.9 months (95% CI 9.1–12.8 months). The survival outcomes of patients with resectable metastasis were more favorable than those of patients with more advanced disease, and the OS time was longer for patients who underwent surgery than for patients who did not undergo surgery during preoperative chemotherapy in Arm B (31.3 months [95% CI 18.9 months–upper level not achieved vs. 15.9 months [95% CI 7.1–22.9 months]). Based on this result, a randomized controlled phase 3 trial is being conducted to investigate the efficacy of surgical resection combined with perioperative FLOT therapy (AIO-FLOT5 trial [24]).

Japanese gastric cancer treatment guidelines also include topics concerning oligometastasis and surgical treatment is weakly recommended for patients with a small number of para-aortic lymph node metastases after neoadjuvant chemotherapy or for patients with solitary liver metastasis. The efficacy of neoadjuvant chemotherapy followed by surgical resection for primary tumors and para-aortic lymph node metastases localized to station No. 16 a2 and station No. 16 b1 [25] were evaluated in the JCOG0405 trial [19]. The phase 2 trial investigated the efficacy and safety of surgical resection after neoadjuvant chemotherapy with S1 and cisplatin (CDDP) therapy for patients with extended lymph node metastasis, including para-aortic lymph node metastasis localized to station No. 16 a2 and station No. 16 b1 and swollen lymph nodes along the celiac artery, splenic artery, common hepatic artery, or left gastric artery. The study included 51 eligible patients, and 49 patients underwent surgery after preoperative chemotherapy. Forty-three patients (87.8%) underwent R0 resection. The 5-year OS rate of patients with para-aortic lymph node metastasis was only 57%, which was remarkably higher than that of historical survival data of patients with para-aortic lymph node metastasis undergoing surgical resection only, in which the 3-year survival rate was as low as 5% [19]. After the JCOG0405 trial, neoadjuvant chemotherapy followed by surgery for primary tumors and lymph node metastasis became the standard treatment strategy for patients with para-aortic lymph node metastasis localized to station No. 16 a2 and station No. 16 b1 [3,25,26]. Solitary liver metastasis is also treated with surgical treatment. A Japanese multicenter retrospective cohort study [22] evaluated the survival outcomes of patients with liver metastasis. The trial included 94 patients from 26 institutions with gastric cancer with synchronous or metachronous liver metastasis. Of these, 69 patients underwent surgical resection, 11 underwent surgery plus microwave coagulation therapy (MCT) or radiofrequency ablation (RFA), and 14 underwent MCT or RFA therapy. Approximately 60% of the patients had solitary liver metastasis, and 10.6% had more than four lesions. Postoperative chemotherapy was applied to 69.1% of the patients. The 3-year OS rate of all patients was 51.4%. This study also investigated the survival outcomes of patients with solitary liver metastasis, and the OS time of these patients was longer than that of patients with multiple liver metastases (HR 2.49, 95% CI 1.41–4.41, *p* = 0.001). Based on this evidence supporting favorable survival outcomes, surgical interventions are weakly recommended to patients with limited para-aortic lymph node metastases or solitary liver metastasis from gastric cancer in Japan.

## 3. OligoMetastatic Esophagogastric Cancer Consortium

According to the current lack of a consensus definition and treatment for oligometastasis of gastric or gastroesophageal cancer, a European multicenter board of experts organized the OligoMetastatic Esophagogastric Cancer (OMEC) Consortium to discuss and establish a definition, diagnosis, and treatment strategy for oligometastasis from gastric cancer or gastroesophageal cancer [27,28,29,30]. The OMEC Consortium consists of 50 esophagogastric cancer expert institutions in Europe, and it has been endorsed by major associations on cancer treatment, such as the European Organisation for Research and Treatment of Cancer, ESMO, and the International Gastric Cancer Association. The OMEC project was conducted, which consists of five chapters, including a systematic review and meta-analysis [28], and it reached a consensus on definitions and treatments for oligometastasis based on case presentations and board meetings [27,29]. According to the OMEC project, oligometastasis is defined as limited metastasis to one organ with a maximum of three lesions or one distant lymph node metastasis in patients with synchronous metastasis. For liver metastasis, two bilobar metastatic lesions or three or fewer unilobar lesions are considered oligometastasis. In addition, three or fewer unilateral lung metastatic lesions, unilateral adrenal gland metastatic lesions, and one metastatic lesion in either the soft tissue or bone are considered oligometastasis. The OMEC project also recommends that the treatment strategy for oligometastasis should involve systemic chemotherapy followed by re-evaluation of the disease condition and local treatment, such as surgical resection or stereotactic radiotherapy. For metachronous oligometastasis in which the interval between the primary tumor and metastasis was within 2 years, upfront local treatment should be considered. After this consensus, the OMEC Consortium announced its plan to conduct a prospective randomized controlled trial in the future [27,28,29,30].

## 4. Treatment Strategy for Gastric Cancer Metastasis According to the Affected Organ

### 4.1. Para-Aortic Lymph Node and Distant Lymph Node Metastasis

Para-aortic lymph node metastasis in station No. 16 a2 or station No. 16 b1 is classified as distant metastasis and is recognized as oligometastasis. Neoadjuvant chemotherapy followed by surgical resection is one of the standard treatment options [19]. Distant lymph node metastasis, such as metastasis to the mediastinal lymph node, axillary lymph node, supraclavicular lymph node, and other extra-abdominal lymph nodes, is recognized as inoperable, and palliative chemotherapy is applied to patients with these conditions. Para-aortic lymph node metastasis at station No. 16 a1 and station No. 16 b2 is retroperitoneal [25] and is considered more advanced than metastasis at station No. 16 a2 and station No. 16 b1. Therefore, it is not considered as resectable [31,32]. Nowadays, for initially unresectable metastatic gastric cancer, the treatment strategy is palliative chemotherapy. For patients with a remarkable response to chemotherapy, surgical resection of the primary tumor and metastatic lesions, if they remain, is used with the intention of curing the disease or prolonging the survival time. This approach has received much attention as conversion therapy, and the type of surgery is conversion surgery [33,34]. The CONVO-GC-1 study was a retrospective, international, multicenter cohort study conducted to evaluate the survival outcomes of patients with distant metastasis treated with surgical resection after chemotherapy at 55 institutions in Japan, Korea, and China [35]. The survival outcomes of patients with para-aortic lymph node metastasis at station No. 16 a2/b1 and station No. 16 a1/b2 were compared. The OS of patients with para-aortic lymph node metastasis at station No. 16 a1/b2 was longer than that of patients with metastasis at station No. 16 a2/b1. The MST was 54.3 months (95% CI 32.6 months–upper limit not reached) and 33.5 months (95% CI 29.2–43.3 months), respectively (*p* = 0.044). This paradoxical survival outcome was considered to be a result of the difference in treatment strategy for each type of metastasis. For patients with 16 a2/b1 lymph node metastasis, neoadjuvant chemotherapy followed by surgery for the primary tumor and resection of the para-aortic lymph node were commonly used. Patients who did not respond to chemotherapy underwent surgery. For patients with 16 a1/b2 lymph node metastasis, surgery was performed in patients who presented a remarkable response to chemotherapy as conversion therapy. Selecting patients appropriately based on their response to chemotherapy may be important in the surgical treatment of patients with distant oligometastasis.

### 4.2. Liver Metastasis

Solitary liver metastasis is considered a type of oligometastasis [3,29]; however, whether two or more lesions should be considered as oligometastasis remains controversial. Several recent studies have evaluated the prognostic factors for synchronous or metachronous liver metastasis from gastric cancer treated with surgical resection [21,22,36,37,38,39,40,41] (Table 1). Concerning the number of metastatic lesions, solitary liver metastasis is a common favorable prognostic factor for surgical resection. Studies have utilized various treatment strategies for liver metastasis, such as upfront surgical resection and/or postoperative or perioperative chemotherapy. The presence of two to three lesions is considered a favorable prognostic factor in some reports. However, for two or three metastatic liver lesions, whether surgical resection is the best treatment option is controversial. Shirasu et al. reported the treatment outcomes of patients with two or three synchronous or metachronous metastatic liver lesions in patients with gastric cancer who were treated with palliative chemotherapy compared with those of patients treated with upfront surgical resection [42]. The progression-free survival (PFS) from treatment initiation was better for patients who underwent palliative chemotherapy than for patients who underwent surgical resection first. The median PFS was 26.1 months and 7.9 months (*p* = 0.012), respectively. The OS time also tended to be longer in the palliative chemotherapy group (38.1 vs. 24.8 months, respectively, *p* = 0.146). Moreover, the palliative chemotherapy group included three patients who underwent hepatectomy during palliative chemotherapy after presenting a remarkable response to chemotherapy, and none of them experienced recurrence after surgery. For two or three metastatic liver lesions, surgical resection may not be effective for all patients, and patients who present with a certain response to chemotherapy may be appropriate candidates for local treatment. According to this evidence, one metastatic liver lesion could be categorized as oligometastasis. Two to three metastatic liver lesions may be categorized as oligometastasis, but this remains controversial, so further investigation is needed to clarify this issue.

### 4.3. Peritoneal Metastasis and Peritoneal Lavage Cytology-Positive Disease

Peritoneal dissemination is the typical metastatic pattern in gastric cancer, for which it is difficult to apply curative resection after chemotherapy [33]. Peritoneal lavage cytology-positive (CY1) disease is classified as distant metastasis in the Japanese classification of gastric carcinoma [25] and is considered the preliminary status of peritoneal dissemination and peritoneal recurrence after gastrectomy for CY1 disease [43]. The standard treatment for patients with CY1 disease has not been established; however, two treatment strategies can be practically applied to patients with gastric cancer with CY1 disease. The first is gastrectomy, followed by postoperative chemotherapy. The survival outcomes of this strategy have been evaluated in retrospective cohort studies, and the 5-year OS of patients with CY1 disease without gross peritoneal dissemination who underwent gastrectomy followed by postoperative chemotherapy with S-1 monotherapy was 26.0–30.2% [44,45]. This result was acceptable, considering that CY1 disease is categorized as distant metastasis. The second approach is surgery if the lavage cytology is negative upon re-evaluation of peritoneal disease during systemic chemotherapy. This treatment strategy is called conversion therapy, and selecting patients with a remarkable response to chemotherapy is useful to determine those that could benefit most from surgical resection of the primary tumor and distant metastasis if they remain after chemotherapy. Gastrectomy causes a decrease in oral intake and loss of body weight. Total gastrectomy can also cause intolerance to chemotherapy [18], so futile gastrectomy should be avoided. As part of conversion therapy, conversion surgery is used with curative intent. The survival outcomes of patients with CY1 disease treated with conversion therapy were evaluated in another retrospective cohort study [46]. The proportion of patients in whom the peritoneal lavage cytology turned negative during chemotherapy was 40.6%, and the 3-year PFS rate was 76.9% (95% CI 47.8–92.4%). Patients in whom the lavage cytology did not turn negative did not undergo futile surgery and continued chemotherapy.

Peritoneal dissemination is also an indicator of unresectable metastasis; however, patients are potentially eligible for surgical resection if the dissemination in the peritoneal cavity is localized near the stomach, which is classified as P1 disease in the Japanese classification of gastric carcinoma, first edition [47]. Similar to CY1 disease, the standard treatment for patients with peritoneal metastasis localized near the stomach has not been established. However, surgical resection followed by postoperative chemotherapy is one option. A Japanese multicenter retrospective cohort study investigated the survival outcomes of patients with peritoneal metastasis localized near the stomach who underwent surgical resection followed by postoperative chemotherapy [45]. The 5-year OS rate of patients with P1 disease without other distant metastasis who underwent complete resection of the primary tumor and who had dissemination localized near the stomach, followed by S-1 monotherapy, was 17.5–19.9% [45]. These findings suggest that patients with CY1 disease or localized peritoneal dissemination who undergo surgical resection combined with postoperative or perioperative chemotherapy have favorable survival outcomes. However, no standard treatment strategy has been established for such patients, and further improvements in survival outcomes are needed.

### 4.4. Krukenberg Tumors from Gastric Cancer

Krukenberg tumors are ovarian metastases from another primary origin, such as gastric cancer. A standard treatment for Krukenberg tumor from gastric cancer has not been established; however, surgical resection is sometimes used. Several reports have analyzed the survival outcomes and prognostic factors [48,49,50,51,52] of patients who undergo surgical treatment for Krukenberg tumor from gastric cancer. The median OS time of patients who underwent surgical resection of Krukenberg tumor was 14–19 months [48,50,51,52]. The prognostic factors of patients who underwent surgical resection are listed in Table 2. Although metastasectomy for Krukenberg tumor has been reported as a favorable prognostic factor in several studies, those studies were retrospective cohort studies, in which there would be bias in which patients with localized metastasis underwent surgical resection of Krukenberg tumor. Peritoneal carcinomatosis has also been reported as a poor prognostic factor in multiple studies, so metastasectomy for the Krukenberg tumor should be applied to patients in whom the metastases are limited to the ovaries, and peritoneal dissemination is not identified. Perioperative chemotherapy may improve the survival outcomes of those patients; however, the most appropriate regimens and the duration of chemotherapy have not yet been clarified, so further investigations are needed.

## 5. Prospective Study Investigating the Survival Outcomes of Patients with Oligometastasis from Gastric Cancer Who Underwent Surgery Compared with Palliative Chemotherapy

For oligometastasis from gastric cancer, surgical treatment is commonly applied as the pivotal treatment modality combined with systemic chemotherapy. However, there is no evidence from prospective randomized controlled studies to suggest the superiority of surgical resection for oligometastasis in terms of survival outcomes over palliative chemotherapy alone. Two randomized controlled trials are currently ongoing. The AIO-FLOT5 (RENAISSANCE) trial [24] is a phase 3 study comparing perioperative FLOT therapy combined with surgical resection of the primary tumor and metastatic lesions with palliative FLOT therapy. The eligible patients are patients with untreated gastric or gastroesophageal junction cancer with limited or localized metastasis, as noted above for the AIO-FLOT3 trial. Registered patients will be treated with four cycles of FLOT therapy before undergoing re-evaluation, and patients without disease progression will be randomized to additional FLOT therapy or surgical resection, followed by four cycles of FLOT therapy. The primary endpoint is OS. The SURGIGAST trial [53] is another phase 3 randomized controlled trial involving patients with gastric cancer with oligometastasis treated with palliative chemotherapy surgical resection followed by systemic chemotherapy or chemotherapy without surgery. The eligible patients are patients with gastric cancer with oligometastasis who have been treated with at least 2 months of first-line chemotherapy. The population with oligometastasis in this trial is almost the same as in the RENAISSANCE trial. The primary endpoint is OS. These two studies are investigating whether the survival outcomes of surgical resection combined with perioperative FLOT or systemic chemotherapy for metastatic gastric cancer are better than those of palliative chemotherapy. If these trials confirm the superiority of surgical treatment over palliative chemotherapy, perioperative systemic chemotherapy followed by surgical resection of primary and metastatic lesions and postoperative chemotherapy will become standard treatment options for eligible patients with localized metastasis, and the localized metastatic diseases included in these trials will be recognized as oligometastasis from gastric cancer. The results of these studies are eagerly awaited.

## 6. Future Directions and Conclusions

For a long time, there has been no common definition or standard treatment approach for oligometastasis from gastric cancer. However, the concept of oligometastasis is becoming increasingly recognized in gastric and gastroesophageal adenocarcinoma. Localized and resectable diseases without a remnant tumor and diseases that can be controlled using systemic chemotherapy are recognized as oligometastasis. In Western countries, the treatment for oligometastasis is developing toward perioperative systemic chemotherapy combined with surgical resection of primary and metastatic lesions. Surgical resection followed by postoperative adjuvant chemotherapy is used in Japan, especially for solitary liver metastasis or localized peritoneal metastasis with lavage cytology-positive disease. Nevertheless, several clinical questions remain unanswered.

Which metastases are appropriate for surgical resection combined with systemic chemotherapy is one of the important clinical questions. If the metastases are localized, the survival outcomes of patients who undergo multimodal treatment differ among metastatic sites. Another important consideration is the appropriate surgical indication after preoperative chemotherapy. In the RENAISSANCE trial, stable disease and a good clinical response to preoperative chemotherapy are eligibility criteria for surgical resection. However, an insufficient response could lead to unsatisfying survival outcomes. Further analyses are needed to determine the prognostic factors of multimodal treatment for patients with oligometastasis from gastric cancer, especially evidence from prospective randomized controlled trials.

Another clinical question is what the most promising perioperative chemotherapeutic regimen is for oligometastasis from gastric cancer. The RENAISSANCE trial adopted FLOT therapy for perioperative chemotherapy, and the SURGIGAST trial did not prescribe chemotherapeutic regimens. Recently, new chemotherapeutic agents have been developed for metastatic gastric cancer. For HER2-positive disease, chemotherapy plus trastuzumab is one of the recommended treatments, and for HER2-negative disease, chemotherapy or chemotherapy plus nivolumab is recommended according to the degree of programmed death-ligand 1 expression [1,2,3]. Moreover, the efficacy of nivolumab in postoperative chemotherapy after complete resection of oligometastasis is still unclear. The ATTRACTION-5 trial was a phase 3 randomized controlled study investigating the efficacy of adjuvant chemotherapy with S-1 or CapeOX plus nivolumab or placebo for resected pStage III gastric cancer. The primary endpoint was recurrence-free survival (RFS). The RFS rate was 68.4% (95% CI 63.0–73.2%) for patients treated with nivolumab combined with adjuvant chemotherapy and 65.3% (95% CI 59.9–70.2%) for patients treated with placebo combined with adjuvant chemotherapy, but the efficacy of adding nivolumab to adjuvant chemotherapy was not confirmed (HR 0.90, 95.72% CI 0.69–1.18, *p* = 0.4363) (ASCO2023, unpublished). The efficacy of nivolumab in postoperative adjuvant chemotherapy has not been investigated in oligometastasis, and further analyses are needed.

Analyses of prognostic factors are also important. The identification of poor prognostic factors for the local treatment of oligometastasis will provide useful information for the development of postoperative chemotherapy. The correlation between survival outcomes and pathological response or complete resection in patients with advanced gastric cancer treated with multimodal treatment has been reported [35]. Molecular residual disease evaluated using circulating tumor DNA (ctDNA) may be another promising prognostic measure for recurrence after the surgical resection of oligometastasis. Although there is little evidence for correlations between survival outcomes and ctDNA in resectable gastric cancer, being positive for ctDNA in resectable colorectal cancer confirmed using liquid biopsy is a robust poor prognostic factor for postoperative recurrence of disease [54]. The usefulness of ctDNA in gastric cancer, including oligometastasis, remains to be investigated in the future. Further discussions and analyses are needed to establish a standard treatment for oligometastasis from gastric cancer.

In conclusion, oligometastasis from gastric cancer indicates localized metastatic disease, which is technically resectable at diagnosis and presents a certain response to preoperative chemotherapy. The efficacy of surgery or multimodal treatment (in which surgery is combined with systemic chemotherapy) for those with limited metastasis has been evaluated for solitary liver metastasis and para-aortic lymph node metastasis in station No. 16 a2 and station No. 16 b1. Ongoing prospective studies will reveal whether perioperative chemotherapy combined with surgery prolongs the survival of patients with other localized metastases in the future. The development of new treatment strategies, including new chemotherapeutic agents and regimens, could improve the survival outcomes of patients with oligometastasis from gastric cancer.

## Figures and Tables

**Table 1 cancers-16-00673-t001:** Number of liver metastases for a favorable prognosis with surgical resection of oligometastasis from gastric cancer.

Authors	Year	Reference	Number of Liver Metastases for a Favorable Prognosis
Granieri S et al.	2021	[41]	1
Montagnani F et al.	2018	[40]	≤3
Oki E et al.	2016	[22]	1
Markar SR et al.	2016	[39]	1
Kinoshita T et al.	2015	[21]	≤2
Liu Q et al.	2015	[36]	1
Petrelli F et al.	2015	[38]	1
Ohkura Y et al.	2015	[37]	≤3

**Table 2 cancers-16-00673-t002:** Prognostic factors of Krukenberg tumor from gastric cancer treated with surgery.

Authors	Year	Reference	Favorable Prognostic Factors	Poor Prognostic Factors
Lin X et al.	2022	[52]	Oophorectomy Fibrinogen ≤ 4 g/L Chemotherapy after ovarian metastasis	Large ovarian metastasis Peritoneal metastasis
Ma F et al.	2019	[51]	Metastasectomy Chemotherapy	Linitis plastica Ascites
Yu P et al.	2017	[50]	Metastasectomy ER-positive PR-positive	Peritoneal carcinomatosis
Cho JH et al.	2015	[48]	Metastasectomy	Signet-ring cells Peritoneal carcinomatosis

ER: estrogen receptor, PR: progesterone receptor.

## Data Availability

No new data were created.

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
