# Peer review of "Oligometastasis of Gastric Cancer: A Review"

_cancers, 2024, doi:10.3390/cancers16030673_

Round 1
Reviewer 1 Report
Comments and Suggestions for Authors
Reviewers' comments:
This article provides a comprehensive review of the treatment of oligometastasis in gastric cancer. The current status of treatment for oligometastasis of gastric cancer is thoroughly examined, and the content of this article is highly valuable for readers. However, some minor revisions are necessary for approval. The comments and suggestions of this reviewer are listed below.
Minor Revisions:
1. In the "Simple Summary" section, please correct "Stomach cancer" to "gastric cancer" to align with the terminology used in the Abstract and the main text.
2.In the Abstract section, on Page 1, Line 32, it is mentioned that the Japan Clinical Oncology Group is conducting a prospective study of multimodal treatment for gastric cancer with oligometastasis. However, this information is not reflected in the paper.
3. In the main text, on Pages 5 to 8, specifically in the "Treatment Strategy for Gastric Cancer Metastasis According to the Affected Organ" section, the description of the section by affected organ is inconsistent and confusing. Kindly revise and ensure consistency in the descriptions.
4. In the main text, on Page 4 Line 141, Page 7 Lines 295 and 321, and Page 9 Line 406, please correct the description of S1 to S-1 for accuracy and consistency.
5. In the main text, on Page 8, regarding the SURGIGAST trial, the text should cite the test referral site, etc. Please make the necessary citation to provide complete information on the SURGIGAST trial.
Please address these minor revisions to enhance the clarity and accuracy of the article. Thank you for your attention to these details.
Author Response
- In the "Simple Summary" section, please correct "Stomach cancer" to "gastric cancer" to align with the terminology used in the Abstract and the main text.
Thank you for your comment. We have changed the stomach cancer to gastric cancer.
- In the Abstract section, on Page 1, Line 32, it is mentioned that the Japan Clinical Oncology Group is conducting a prospective study of multimodal treatment for gastric cancer with oligometastasis. However, this information is not reflected in the paper.
Thank you for your comment. This JCOG study indicated JCOG0405 study, however, the object of this study is not only patients with oligometastasis, but patients with extensive lymph node metastasis were also included. We remove the sentence from manuscript because it is confusing.
- In the main text, on Pages 5 to 8, specifically in the "Treatment Strategy for Gastric Cancer Metastasis According to the Affected Organ" section, the description of the section by affected organ is inconsistent and confusing. Kindly revise and ensure consistency in the descriptions.
Thank you for your comment. We have revised the sentences to “Liver metastasis”.
- In the main text, on Page 4 Line 141, Page 7 Lines 295 and 321, and Page 9 Line 406, please correct the description of S1 to S-1 for accuracy and consistency.
Thank you for your comment. Re have revised and unified the expression.
- In the main text, on Page 8, regarding the SURGIGAST trial, the text should cite the test referral site, etc. Please make the necessary citation to provide complete information on the SURGIGAST trial.
Thank you for your comment. We have inserted the citation.
Reviewer 2 Report
Comments and Suggestions for Authors
I think the article is interesting for the wide literature revision and for the reasonable conclusions in the field of the treatment of this not-rare advanced GC.
The article is not news, since its nature of review but the argument has been treated completely and exhaustively, so it could be of some interest.
- The chosen topic is of some interest due to the current ack of definitive guidelines in the treatment of oligometastatic GC. The aim of this review is to examine the newest scientific literature trying to answer about the best multimodal treatment in this field.
- Even if the topic is not original, because many studies have been published in the recent past, nevertheless I believe the authors have been able to clearly define the concept of Oligometastatic Gastric Cancer, in which different therapeutic options are possible today
- The article definitely doesn't add anything to the current knowledge, but it has the quality to be updated and unbiased in the information reported
- The debate remains open, but from this review rises a kind of conclusion about the effectiveness of surgery in the treatment of this specific stage of gastric cancer
- The debate remains open, but from this review rises a kind of conclusion about the effectiveness of surgery in the treatment of this specific stage of gastric cancer
- References are appropriate
- There are 2 tables that only enhance what is already reported in the text, concerning the analyzed articles and the number of cases. They are of no particular interest
Author Response
- The chosen topic is of some interest due to the current ack of definitive guidelines in the treatment of oligometastatic GC. The aim of this review is to examine the newest scientific literature trying to answer about the best multimodal treatment in this field.
Thank you for your comment. Oligometastasis is now one of topics in metastatic gastric cancer.
- Even if the topic is not original, because many studies have been published in the recent past, nevertheless I believe the authors have been able to clearly define the concept of Oligometastatic Gastric Cancer, in which different therapeutic options are possible today
Thank you for your comment. We have reviewed the current concept of oligometastasis in gastric cancer.
- The article definitely doesn't add anything to the current knowledge, but it has the quality to be updated and unbiased in the information reported
Thank you for your comment. The reports which were included to this manuscript were important and insightful ones and are helpful for readers to cultivate deeper understanding.
- The debate remains open, but from this review rises a kind of conclusion about the effectiveness of surgery in the treatment of this specific stage of gastric cancer
Thank you for your kind review of this manuscript.
- The debate remains open, but from this review rises a kind of conclusion about the effectiveness of surgery in the treatment of this specific stage of gastric cancer
Thank you for your kind review of this manuscript.
- References are appropriate
Thank you for your kind review of this manuscript.
- There are 2 tables that only enhance what is already reported in the text, concerning the analyzed articles and the number of cases. They are of no particular interest
Thank you for your kind review of this manuscript.